# MTA, an RNA m^6^A Methyltransferase, Enhances Drought Tolerance by Regulating the Development of Trichomes and Roots in Poplar

**DOI:** 10.3390/ijms21072462

**Published:** 2020-04-02

**Authors:** Liang Lu, Yan Zhang, Qizouhong He, Zengxing Qi, Geng Zhang, Wenchao Xu, Tao Yi, Gangning Wu, Ruili Li

**Affiliations:** 1Beijing Advanced Innovation Center for Tree Breeding by Molecular Design, Beijing Forestry University, Beijing 100083, China; 2College of Biological Sciences and Technology, Beijing Forestry University, Beijing 100083, China

**Keywords:** N6-methyladenosine, methyltransferase, trichomes, root system, drought tolerance

## Abstract

N6-methyladenosine (m^6^A) is the most prevalent internal modification present in the mRNAs of all higher eukaryotes, where it is present within both coding and noncoding regions. In mammals, methylation requires the catalysis of a multicomponent m^6^A methyltransferase complex. Proposed biological functions for m^6^A modification include pre-mRNA splicing, RNA stability, cell fate regulation, and embryonic development. However, few studies have been conducted on m^6^A modification in trees. In particular, the regulation mechanism of RNA m^6^A in *Populus* development remains to be further elucidated. Here, we show that PtrMTA (*Populus trichocarpa* methyltransferase) was colocalized with PtrFIP37 in the nucleus. Importantly, the *PtrMTA*-overexpressing plants significantly increased the density of trichomes and exhibited a more developed root system than that of wild-type controls. Moreover, we found that *PtrMTA*-overexpressing plants had better tolerance to drought stress. We also found *PtrMTA* was a component of the m^6^A methyltransferase complex, which participated in the formation of m^6^A methylation in poplar. Taken together, these results demonstrate that *PtrMTA* is involved in drought resistance by affecting the development of trichomes and roots, which will provide new clues for the study of RNA m^6^A modification and expand our understanding of the epigenetic molecular mechanism in woody plants.

## 1. Introduction

More than 150 nucleotide modifications of RNA have been described, and more than 10 of them have been reported in mRNA [1]. RNA methylation, especially N6-methyladenosine (m^6^A), is an abundant internal modification of mRNAs in eukaryotes [2,3]. N6-methyladenosine was first discovered in bacteria DNA in 1955 [3]. Since the discovery of m^6^A methylation, m^6^A has been found in viral RNAs and various other eukaryotes, including mammals [4,5,6,7,8], yeast [9], fruit flies [10], and plants, such as oats [11], maize [12], *Arabidopsis* [13], and wheat [14]. m^6^A modifications are mainly distributed in mRNA and long non-coding RNA. In addition, m^6^A also exists in other types of eukaryotic RNA, such as tRNA (transport RNA) [15], rRNA (ribosomal RNA) [16], and snRNA (small nuclear RNA) [17].

In mammals, m^6^A mRNA methylation is catalyzed by the methyltransferase complex containing METTL3 (methyltransferase 3) [17], METTL14 (methyltransferase 14) [18] and WTAP (Wilms’ tumor 1-related protein) [19,20]. Methyltransferases may be conserved in eukaryotes [21,22]. Recently, it has been found that KIAA1429 may be a new element of the m^6^A methyltransferase complex, which may be related to the formation of m^6^A [21]. The m^6^A modification can be demethylated by m^6^A demethylases, such as fat and obesity-related protein (FTO) [23] and its homolog alkylated DNA repair protein ALkB homolog 5 (ALKBH5) [24]. Therefore, m^6^A modification is a reversible biological process.

The m^6^A-modified transcripts can be recognized by “reader” proteins, such as the YTH (YT512-B homology) domain protein family in animals [25,26,27,28]. To date, 13 proteins with YTH domains have been reported in *Arabidopsis*, including ECT1-11 (Evolutionarily Conserved C-Terminal Region1-11) [25,26,27,28]. Recently, studies have shown that ECT2 (Evolutionarily Conserved C-Terminal Region2) can regulate trichome morphology by affecting the stability of mRNA and plays an important role in trichome branching in *Arabidopsis* [29,30]. In addition, ECT2 can control the developmental timing of leaf formation in *Arabidopsis* [31]. More importantly, the m^6^A-modified transcripts appear to be involved in a wide range of biological processes, including mRNA translatability and stability [22,26,27,32,33], mRNA export from the nucleus [34], alternative polyadenylate site selection [35], splicing regulation [36,37,38], and other mechanisms that accompany RNA maturation [2,3].

At present, the functional studies of methyltransferase plant homologs are mainly concentrated on the herbaceous model plant—*Arabidopsis thaliana*. Luo et al. found that m^6^A methylation was highly conserved and abundant in mRNA modification by comparing the m^6^A methylomes of two different *Arabidopsis* lines [39]. *Arabidopsis* MTA (N6-adenosine-methyltransferase MT-A70-like, a homolog of human METTL3) is expressed at relatively low levels in most tissues but is highly expressed in flowers, meristems, seeds, and microspores [13]. In addition, *AtMTA* also affects the development of embryos and trichomes and is required for m^6^A mRNA methylation [13]. Embryos cannot develop from the globular to the heart stage when *AtMTA* is knocked out in *Arabidopsis* [13]. FIP37 (FKBP12 interacting protein 37 KD) has been found to be a homolog of mammalian WTAP in *Arabidopsis*, and embryo development in the *fip37* mutant stopped at the globular stage [40]. Recently, Shen et al. found that *AtFIP37* can regulate the shoot stem cell fate in *Arabidopsis* [41]. *AtFIP37*-overexpressing plants showed increased trichome branches, suggesting that *AtFIP37* affects the development of trichomes [40]. Plant trichomes play an important role in buffering environmental and plant interactions [42]. In addition, plant trichomes have biological functions, such as reducing transpiration and drought resistance [42,43,44].

Although functional studies of MTA have been reported in herbaceous plants, its potential functions in ligneous plants, especially in trees, remain unknown. Poplar is the dominant species of forest ecosystems, which possesses great economic and ecological value [45]. Poplar has become a model ligneous plant to explore gene function because of its fast growth and other advantages. In this study, we demonstrated that *Populus trichocarpa* methyltransferase (PtrMTA) and PtrFIP37 (FKBP12 interacting protein 37 KD form *Populus trichocarpa*) were co-localized in the nucleus. Moreover, we revealed *PtrMTA*-overexpressing plants had a higher density of trichomes and a more developed root system. In addition, *PtrMTA* enhanced drought resistance in poplar. We also revealed that *PtrMTA* participated in the formation of m^6^A methylation in 84K poplar. In conclusion, we found that *PtrMTA* plays an indispensable role in the growth and resistance of poplar, which provides a basis for systematic research on m^6^A in the future.

## 2. Results

### 2.1. Cloning and Obtaining Transgenic Plants of PtrMTA

To elucidate the potential function of *PtrMTA* in ligneous plants, we constructed a phylogenetic tree of *MTA* using DNAMAN software. We found that *PtrMTA* (Potri.001G085200.1) had 99% homology with *PagMTA* and *PeMTA* (Appendix A). We further found that MTA was highly conserved among a variety of plant species through multiple sequence alignments, such as *Populus trichocarpa*, 84K poplar, *Populus euphratica*, *Oryza sativa*, *Prunus persica*, *Glycine max*, *Solanum lycopersicum*, *Fragaria vesca,* and *Arabidopsis thaliana*. (Appendix A).

To further study the potential biological functions of *PtrMTA*, we cloned the *MTA* gene from *Populus trichocarpa* and constructed a 35S overexpression vector. Next, we performed genetic transformation through the leaf disc method [46] (Figure 1A). The leaves can differentiate into adventitious buds under light culture conditions (Figure 1B). Through screening for resistance, we initially obtained *PtrMTA-*overexpressing transgenic plants (Figure 1C). Every transgenic line was verified by PCR (polymerase chain reaction) and qRT-PCR (quantitative real-time PCR) (Figure 1D, E). In the literature, these two methods are usually used to verify transgenic plants. Through the verification of the two methods, we obtained transgenic poplar 84K (*P. alba* X *P. glandulosa*), i.e., 14 lines that overexpressed *PtrMTA*.

### 2.2. Subcellular Localization of PtrMTA

Previous studies indicated that *Arabidopsis* MTA, an m^6^A methyltransferase, was localized in the nucleus [13]. Therefore, PtrMTA, a homolog of MELLT3, might localize to the same cellular structures. To determine the subcellular localization of PtrMTA, we constructed a *35S::PtrMTA-GFP* fusion protein (green fluorescent protein) expression vector and transiently transformed it into tobacco leaves. Using laser scanning confocal microscopy (LSCM), we found that PtrMTA-GFP was localized in the nucleus (Figure 2A). We further transiently transformed tobacco leaves with the *35S::PtrMTA-GFP* fusion protein and *35S::PtrFIP37-mCherry* fusion protein. LSCM observation showed that *35S::PtrMTA-GFP* and *35S:: PtrFIP37-mCherry* were co-localized in the nucleus (Figure 2B).

### 2.3. Overexpression of PtrMTA Affected Trichome Development

To determine whether *PtrMTA* affected the development of trichomes in poplar, we selected *PtrMTA*-overexpressing plants (OE-PtrMTA-14, OE-PtrMTA-10, OE-PtrMTA-6) and wild type (WT) plants for observation of leaf trichomes. Using scanning electron microscopy (SEM), we found that the trichomes of poplar were single-celled, non-glandular trichomes that did not have branches (Figure 3A). Furthermore, we counted the number of trichomes in the 20 areas (1500 μm * 1000 μm) of leaves in each line. In the transgenic line OE-PtrMTA-14, the number of trichomes was about 13, which was slightly more than in the WT (Figure 3B). The number of trichomes of transgenic line OE-PtrMTA-10 was about 18 (Figure 3C). Compared with the WT, the number of trichomes in the transgenic line OE-PtrMTA-6 was significantly increased and was about two times that of the WT (Figure 3D). More significantly, through statistical analysis, we found the number of trichomes in the *PtrMTA*-overexpressing plants was significantly greater than that in the WT (Figure 3E and Appendix A). Our results illustrated that overexpression of *PtrMTA* can increase the density of trichomes in poplar.

### 2.4. PtrMTA-Overexpressing Poplar had a More Developed Root System

To investigate whether *PtrMTA* affected root development in poplar, we selected WT, OE-PtrMTA-14, OE-PtrMTA-10, and OE-PtrMTA-6 seedlings aged 20 days and potted the seedlings for 50 days. At the seedling stage, the *PtrMTA*-overexpressing plants showed a more developed root system than that of the WT (Figure 4A). The root length of *PtrMTA*-overexpressing plants was significantly longer than that of the WT (Figure 4B). In addition, the number of root tips in the *PtrMTA*-overexpressing plants was about five, which was also significantly higher than that of the WT (Figure 4C). Moreover, we examined the *PtrMTA-*overexpressing poplars that had been grown for 50 days and found that their root systems were significantly more developed than those of the WT (Figure 4D). Compared with the WT roots, the root length of *PtrMTA*-overexpressing plants (about 10 cm) was longer (Figure 4E). The root fresh and dry weights of different lines were measured. We found that the root weight of *PtrMTA*-overexpressing plants was significantly higher than that of the WT (Figure 4F). In summary, *PtrMTA*-overexpressing plants had a more developed root system compared with the WT.

### 2.5. OE-PtrMTA Poplar Possessed Drought Tolerance

We further analyzed the approximately 3000 bp promoter sequence upstream of the PtrMTA ATG using the https://www.ncbi.nlm.nih.gov/pubmed/11752327 website. Interestingly, we found that the *PtrMTA* promoter contained MYB binding sites involved in drought induction and cis-acting elements involved in stress responses (Appendix A). Furthermore, to investigate whether *PtrMTA* increased the drought resistance of poplar, *PtrMTA-*overexpressing poplar and WT poplar plants were cultured under the same conditions and were then subjected to drought treatment to reduce the soil RWC (relative water content) from 70%. On day 2, some leaves of the WT poplars were slightly withered, while the *PtrMTA-*overexpressing (OE-PtrMTA) poplars appeared normal (Figure 5A). On day 5, the leaves of the WT poplars were more seriously withered than those of the *PtrMTA-*overexpressing poplars (Figure 5A).

As the malondialdehyde (MDA) content is an effective indicator of the oxidative damage of the cytomembrane, we measured the changes in the MDA content in leaves of WT and *PtrMTA-*overexpressing poplars under drought stress. The results showed that the MDA content of OE-PtrMTA poplars was lower than that of WT poplar under drought stress (Figure 5B). Superoxide dismutase (SOD) is an important antioxidant that removes reactive oxygen species. Therefore, we tested the activities of SOD in OE-PtrMTA and WT poplars before and after drought treatment. The results showed that the activity of SOD was significantly higher in OE-PtrMTA poplars than in the WT under drought stress (Figure 5C). Except that the soil RWC remained at 70%, the control plants were maintained under the same culture conditions, and no significant differences in stem diameter and plant height were found between OE-PtrMTA poplars and the WT (Appendix A). Therefore, we can conclude that OE-PtrMTA transgenic poplars had better tolerance to drought stress.

### 2.6. PtrMTA Affected the Level of m^6^A

To study whether *PtrMTA* affected the level of m^6^A, we selected WT, OE-PtrMTA-14, OE-PtrMTA-10, and OE-PtrMTA-6 seedlings aged 30 days and extracted RNA to measure the level of m^6^A. Compared to the WT, the level of m^6^A was significantly higher in *PtrMTA*-overexpressing plants (Figure 6A). Furthermore, we also detected the m^6^A level in the roots, stems, and leaves of the WT and overexpressing transgenic seedlings. Importantly, we found that the level of m^6^A in the roots of *PtrMTA*-overexpressing plants was higher than that of the WT, and the m^6^A content in the roots of OE-PtrMTA-6 was about twice as high as that of the WT (Figure 6B). It is worth mentioning that the level of m^6^A in stems of *PtrMTA*-overexpressing plants was much higher than that of the WT (Figure 6C). In addition, the m^6^A content in leaves was lower than that in the roots and stems, but the m^6^A content in *PtrMTA*-overexpressing plants was still significantly higher than that in the WT (Figure 6D). These results indicate that *PtrMTA* is a component of m^6^A methyltransferase and affects the level of m^6^A.

## 3. Discussion

RNA m^6^A is a conserved and abundant mRNA modification that affects many aspects of RNA metabolism [34]. In 2015, Chen et al. found that when the level of m^6^A modification was lowered. It was found to impede cell reprogramming seriously [47]. In mammals, loss of function in the core component of the m^6^A methyltransferase complex, including METTL3 and WTAP, leads to embryonic death and reduces the levels of m^6^A [48,49]. In *Arabidopsis*, AtMTA (a homolog of METTL3) and AtFIP37 (a homolog of WTAP) are localized in the nucleus [13]. Moreover, when the fluorescent fusion proteins MTA-YFP and FIP37-CFP were transiently transformed into *Allium cepa* epidermal cells, it was found that the two proteins were localized to the same nuclear position [13]. Therefore, AtMTA and AtFIP37 were co-localized in the nucleus. Importantly, AtMTA and AtFIP37 are critical for early embryonic development [13,40,41]. T-DNA knockout of MTA or FIP37 led to the stagnation of embryonic development during the global stage and also reduced levels of m^6^A in different tissues [13,40]. In *Arabidopsis*, the demethylases include an AlkB family of 13 homologous proteins [29,50,51]. ALKBH9B participates in defense against viral infections [50]. ALKBH10B is an mRNA m^6^A demethylase that affects vegetative growth and floral transition [51]. In our study, we transiently transformed the *35S::PtrMTA-GFP* fusion protein into tobacco leaves and found that PtrMTA targeted the nuclear spots. In addition, we simultaneously transferred *35S::PtrMTA-GFP* and *35S::PtrFIP37-mCherry* into tobacco leaves and found that the two proteins were co-localized in the nucleus. In addition, we found that the level of m^6^A in *PtrMTA-*overexpressing plants increased significantly compared to the WT. This indicated that PtrMTA is a component of m^6^A methyltransferase and can affect the level of m^6^A.

Trichomes and roots are often associated with plant resistance [52]. A trichome is a hairy structural appendage extending from the epidermal tissues of aerial parts; trichomes include glandular and non-glandular hairs, linear trichomes, and branched trichomes [53]. Trichomes have a variety of biological functions, such as reducing transpiration. Moreover, trichomes are involved in defense against adverse conditions, such as cold, drought, high temperature, strong radiation, as well as disease infestation [53]. Glandular hairs can secrete chemicals, such as alkaloids, aromatic oils, and resins, to defend against biotic and abiotic stresses [53]. Non-glandular hairs play an important role in resisting drought and mechanical damage [54]. The root system is the organ through which a plant directly senses water signals from the soil and absorbs soil water [55]. The root length and depth directly affect the absorption and utilization of soil water and nutrients [56]. In addition, the root tip number affects the range of the root system depth and breadth in space and also affects the ability to absorb water [57]. Root length, root tip number, and root weight are all closely related to plant drought resistance [58]. In this study, the results of SEM showed that overexpression of *PtrMTA* significantly increased the density of trichomes in poplar. In addition, we found that the root length, root tip number, and root weight of *PtrMTA*-overexpressing plants were significantly increased compared to WT plants. These results indicate that *PtrMTA* is closely related to the development of trichomes and plant roots. More importantly, we found that *PtrMTA*-overexpressing plants had stronger drought resistance than the WT under drought treatment. We speculate that *PtrMTA* may be involved in the regulation of drought resistance.

The balance between the generation and elimination of reactive oxygen species in a cell would be broken under stress conditions [59]. Excess reactive oxygen species first attack the cell membrane system, thus destroying the cell membrane [60,61]. MDA is the final product of cytoplasmic membrane peroxidation. MDA can combine with proteins on the cell membrane to inactivate it, thereby destroying the structure and function of the biological membranes [60]. Plants have a membrane protection system that can remove excess reactive oxygen species from cells [60,62]. This membrane protective system is an antioxidant system composed of many enzymes, such as SOD [62]. SOD can protect the biological membrane system by removing harmful reactive oxygen species [63]. Therefore, the MDA content and SOD activity can reflect the resistance of plants to a certain extent [59]. In this study, we found that *PtrMTA*-overexpressing plants were more resistant to drought stress compared to WT plants. Therefore, we measured the MDA content and SOD activities of *PtrMTA-*overexpressing plants before and after drought treatment. After drought treatment, the MDA content was significantly lower than that of the WT, which may be related to the strong drought tolerance of *PtrMTA*-overexpressing plants. In addition, the SOD activity of *PtrMTA-*overexpressing plants was obviously higher than that of the WT, which may be related to resistance to stress.

## 4. Materials and Methods

### 4.1. Plant Materials and Plant Growth Conditions

In this study, the experimental materials—Tobacco, *Populus trichocarpa,* and 84K poplar—were planted in a greenhouse of the Beijing Forestry University at 25°C under a 16 h light (240 μmol m ^–2^ s ^–1^) and 8 h dark. Potted *Populus trichocarpa* were watered based on evaporation demand and were regularly irrigated with water.

### 4.2. cDNA Cloning and Plasmid Construction

Whole seedlings of poplar were collected and stored in liquid nitrogen. The total RNA was isolated using an RNA Isolation Kit (TIANGEN, Beijing, China). In the last step, potentially contaminating DNA was removed by treatment with DNase I. Four micrograms of total RNA was used for the reverse transcription reaction with a First-Strand cDNA Synthesis Kit (TIANGEN, Beijing, China) according to the protocol. The cDNA of *PtrMTA* and *PtrFIP37* was cloned by PCR. The DNA polymerase we used was ex-Taq (Takara, Japan). The initial denaturing time was 95 °C for 5 min, followed by 35 cycles at 95 °C for 30 s, 56 °C for 30 s and 72 °C for 2.5 min, with a final extension at 72 °C for 5 min. Furthermore, the PtrMTA sequence was inserted between the KpnI (Takara, Japan) and BamHI (Takara, Japan) sites of the *pCAMBIA-2301* and *pCAMBIA-2301-GFP* vectors. We transformed the constructed *pCAMBIA-2301-MTA* and *pCAMBIA-2301-MTA-GFP* vectors into *Agrobacterium tumefaciens* GV3101. The transformed *Agrobacterium* was then applied to Luria-Bertani (LB) solid plates with 50 mg/L kanamycin and 50 mg/L rifampicin antibiotics and incubated at 28 °C for 2–3 days. A single clone was picked and subsequently verified by PCR. The PtrFIP37 sequence was inserted between the KpnI (Takara, Japan) and XbaI (Takara, Japan) sites of the *pCAMBIA-1300-mCherry* vector. *pCAMBIA-1300-FIP37-mCherry Agrobacterium* transformation and PCR were performed using the same method as above. The primers are in Appendix A.

### 4.3. qRT-PCR (Quantitative Real-Time Polymerase Chain Reaction) Analysis

RNA from transgenic plants and WT plants was extracted using an Adelaide RNA Kit (Adelaide, China). The RNA was reverse transcribed into cDNA according to the method provided by the One-Step gDNA Removal and cDNA Synthesis SuperMix (TransGen Biotech, China). qRT-PCR was performed using SYBR Green Mix (Takara, Japan) in an optical 96-well plate. As an internal control, 18S rRNA was used in qRT-PCR. The qRT-PCR instrument we use was Bio-Rad-CFX96 Touch, America. The 2-DDCt analysis method was used to analyze the relative expression of *PtrMTA*. The initial denaturing time was 95 °C for 2 min, followed by 40 cycles at 95 °C for 5 s, 60 °C for 5 s, and 95 °C for 5 s. The primers are in Appendix A.

### 4.4. Phylogenetic and Domain Analyses of PtrMTA

The homologous amino acid sequences of MTA in different plant species were obtained from the NCBI database (https://www.ncbi.nlm.nih.gov/) and Phytozome database (https://phytozome.jgi.doe.gov/pz/portal.html). In NCBI, PtrMTA homologous sequences were obtained by the BLASTn method (Appendix A). In addition, the phylogenetic tree of PtrMTA was simply created using DNAMAN. DNAMAN was used for comparison of PtrMTA amino acid sequences and confirmed conserved structures.

### 4.5. Genetic Transformation of Poplar and Identification of Transgenic Seedlings

A single colony of *Agrobacterium* was shaken in LB medium at 28 °C overnight until the culture density reached an OD_600_ of 0.6–0.8. *Agrobacterium* cells were harvested by centrifugation at 5000 rpm for 5–8 min. The harvested cells were suspended in a ½-strength MS solution (pH 5.8) containing 5% (w/v) sucrose and acetosyringone (100 μM) as a conversion solution (Song et al., 2019). The pretreated leaves were placed in the transformation solution and soaked for about 15 min, during which time shaking was required every 3 min. The infested leaves were placed on sterile filter paper, and extra *Agrobacterium* was aspirated. The leaves were placed in ½-strength MS medium for 3–5 days under dark culture. The dark-treated leaves were placed in a medium containing kanamycin for differentiation culture. When the shoots grew to 1 cm after 4–5 weeks, they were cut off and transferred to a resistant screening medium. The obtained transgenic plants were screened for DNA identification and qRT-PCR identification. The primers are in Appendix A.

### 4.6. Subcellular Localization

The *Agrobacterium* was collected by centrifugation for 5 min at 5000 g. The cultures were resuspended in a medium containing 5 mM MES, 100 mM MgCl_2_ (pH 5.6), and 0.1 mM acetylsyringone [64]. The *Agrobacterium* cells were then injected into the abaxial surfaces of the tobacco leaves by using a needleless syringe. The plants were then cultured in a dark environment for 24 h. The transiently transformed tobacco leaves were fixed in sterile water and imaged with a Leica 20X or 40X water immersion objective, white light laser, and a hybrid detector with a resolution of 1024 × 1024 pixels. All images shown in this article are original images taken from the Leica SP8.

### 4.7. SEM Analysis of Poplar Leaves Trichomes

SEM (Hitachi S4800, Japan) was used to study the trichome density of 40-day-old WT and *PtrMTA*-overexpressing poplar leaves. We selected potted seedlings that had grown for 40 days and picked the sixteenth leaf of each plant. Then we cut out the middle area near the main vein of the leaf. Poplar leaves were trimmed to a size of 1 cm^2^. Samples were then immediately drop fixed in formalin–acetic acid–alcohol (FAA was composed of 5 mL 38% formaldehyde, 5 mL acetic acid and 90 mL 70% alcohol). The samples were dehydrated successively with 30%, 50%, 60%, 70%, 80%, and 90% ethanol for 30–40 min and were dehydrated overnight with anhydrous ethanol. A dryer was used for the dehydrated samples (Leica EM CPD300, Germany). Dry samples were sputter-coated with gold (110 s). Finally, the samples were imaged using a scanning electron microscope. We counted the number of trichomes in the 20 areas (1500 μm * 1000 μm) of leaves in WT and transgenic plants.

### 4.8. Drought-Tolerance Experiments

Three plants of the 3 transgenic lines and one WT poplar plant were grown in a greenhouse at 25 °C during the day and 22 °C at night. The greenhouse was illuminated for 16 h per day. All plants were grown in suitable pots with a diameter of 13.5 cm. All plants were subjected to a short-term drought treatment in which the soil RWC (relative water content) decreased from 70% [65,66]. The control plants were kept under the same conditions.

### 4.9. Measurement of MDA Content and SOD Activity

At 0 and 7 days, leaves that detached from drought-treated plants were measured to determine the MDA content and SOD activity. The MDA content and SOD activity of transgenic and WT plants were determined using a Solarbio Kit (Solarbio, China). MDA (nmol/g FW) = 5*(6.45*(A532–A600)-0.56*A450)/W (W: sample quality); SOD (U/g FW) = 11.4*inhibition percentage/(1-inhibition percentage)/W*F (W: sample quality; F: sample dilution multiple).

### 4.10. Determination of the m^6^A Level in RNA

RNA from transgenic and WT seedlings was extracted using an Adelaide RNA Kit (Adelaide China). The RNA concentrations of different samples were determined using a NanoPhotometer NP80 (Implen, Germany). The m^6^A level was determined using an Aderr kit (Epigentek, America). m^6^A (%) = (SampleOD-NcOD)/S/(PcOD-NcOD)/P*100% (Nc: Negative control; Pc: Positive control; S: RNA sample inputs, ng; P: Positive control inputs, ng).

## 5. Conclusions

This study shows that the m^6^A methyltransferase MTA from *Populus trichocarpa* was localized in the nucleus and co-localized with PtrFIP37 in the nucleus. Moreover, the overexpression of *PtrMTA* significantly affected the level of m^6^A and increased the density of trichomes. *PtrMTA*-overexpressing plants had a more developed root system. More importantly, the overexpression of *PtrMTA* can enhance the drought resistance of poplar. Therefore, our research not only enriched the study of the m^6^A function in *Populus* but also provided a basis for exploring the epigenetic molecular mechanism in woody plants.

## Figures and Tables

**Figure 1 ijms-21-02462-f001:**
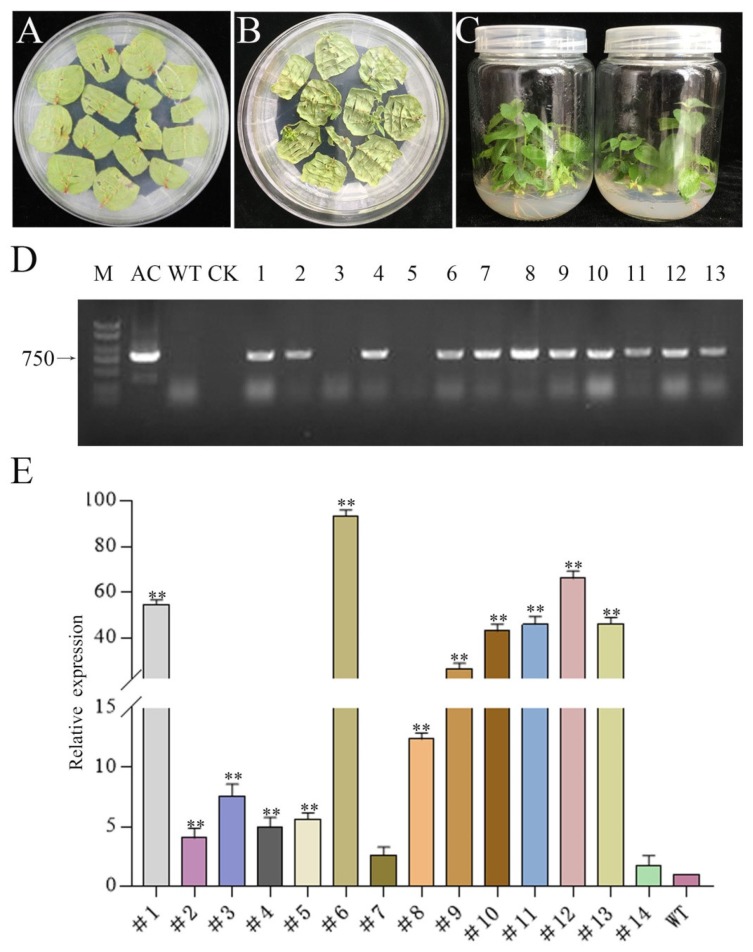
Obtaining and identification of *Populus trichocarpa* methyltransferase *(PtrMTA)* transgenic plants. (**A**–**C**) Genetic transformation of 84K poplar and obtained transgenic seedlings. (**D**) PCR detection of transgenic plants (M: DNA marker 2000^+^; AC: active control; WT: 84K wild type; CK: control check). (**E**) Expression level analysis of PtrMTA transgenic plants by qRT-PCR. Error bars are means ± SE (*n* = 3). All asterisks denote significant differences: ** *p* < 0.01.

**Figure 2 ijms-21-02462-f002:**
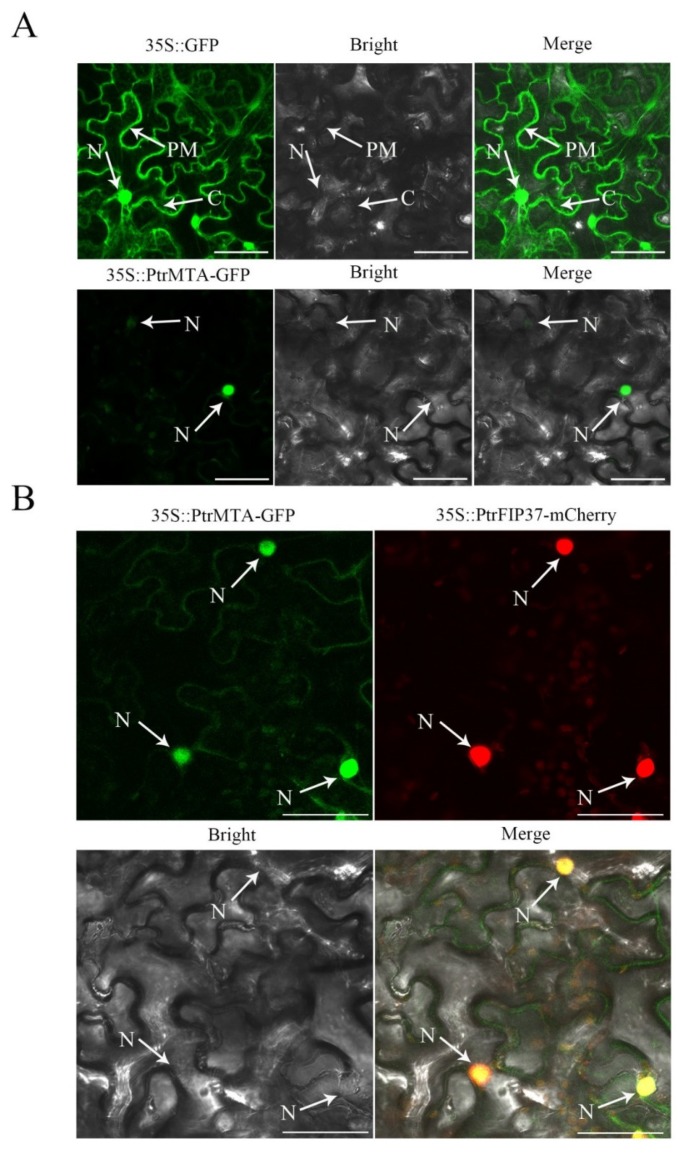
PtrMTA was targeted to the nucleus. (**A**) Subcellular localization of *35S::GFP* and *35S::PtrMTA-GFP* in transiently expressed tobacco leaves. Bar = 15 μm. (**B**) Subcellular co-localization of *35S::PtrMTA-GFP* and *35S::PtrFIP37-mCherry* in transiently expressed tobacco leaves. Bright: bright-field image; PM: plasma membrane; C: cytoplasm; N: nucleus. Bars = 18 μm.

**Figure 3 ijms-21-02462-f003:**
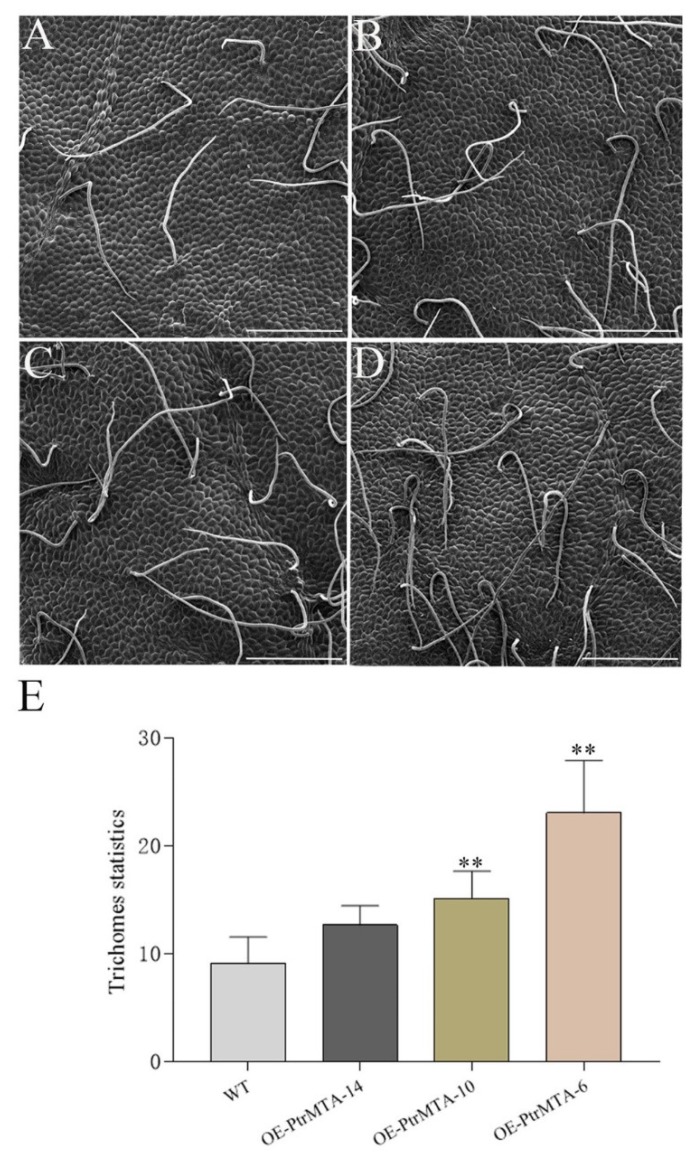
PtrMTA increases trichome density in transgenic 84K poplar plants. (**A**) Illustration of trichome density in wild type (WT) plants. (**B**–**D**) Illustration of the increase in trichome density in *35S::PtrMTA* lines OE-PtrMTA-14, OE-PtrMTA-10, and OE-PtrMTA-6, respectively. (**E**) Statistics on the number of trichomes in the 20 areas (1500 μm * 1000 μm) of WT and transgenic 84K poplar leaves. Bars = 200 μm. Error bars are means ± SE (*n* = 20). All asterisks denote significant differences: ***p* < 0.01.

**Figure 4 ijms-21-02462-f004:**
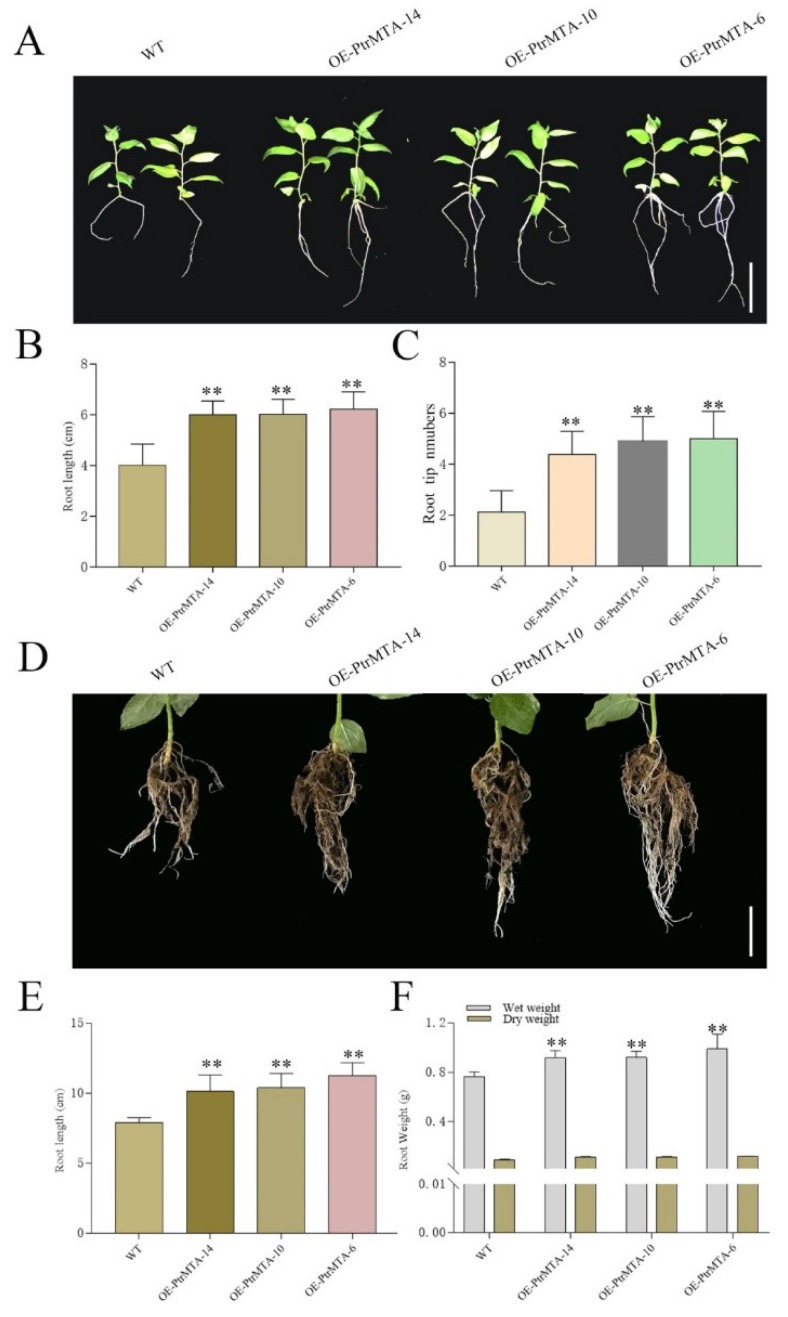
Root length, root tip number, and root weight of the WT and *PtrMTA*-overexpressing plants were compared. (**A**) Morphological differences between WT and transgenic plants grown for 20 days. (**B**) Comparison of root lengths between wild-type and *PtrMTA-*overexpressing plants ogrown for 20 days. (**C**) Comparison of root tip numbers between wild-type and transgenic plants grown for 20 days. (**D**) Morphological differences between wild-type and transgenic plants grown for 50 days in pots. (**E**) Comparison of root lengths between wild type and *PtrMTA-*overexpressing plants grown for 50 days. (**F**) Comparison of root fresh weight and dry weight in WT and transgenic plants. Bars = 3 cm, A; 5 cm, D. Error bars are means ± SE (*n* = 10). All asterisks denote significant differences: ***p* < 0.01.

**Figure 5 ijms-21-02462-f005:**
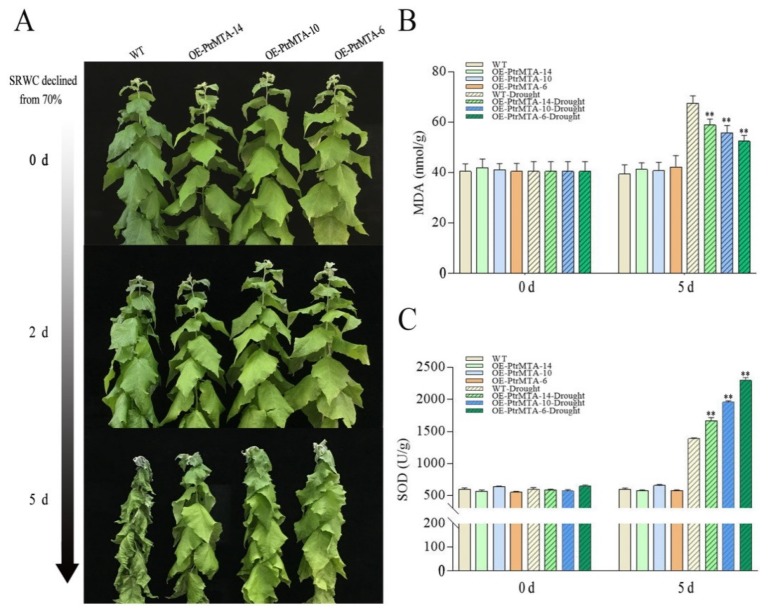
OE-PtrMTA plants had increased tolerance under drought treatment. (**A**) Phenotypic differences in drought conditions. (**B**) Analysis of the malondialdehyde (MDA) content. (**C**) Superoxide dismutase (SOD) activity determination. Error bars are means ± SE (*n* = 4). All asterisks denote significant differences: ***p* < 0.01.

**Figure 6 ijms-21-02462-f006:**
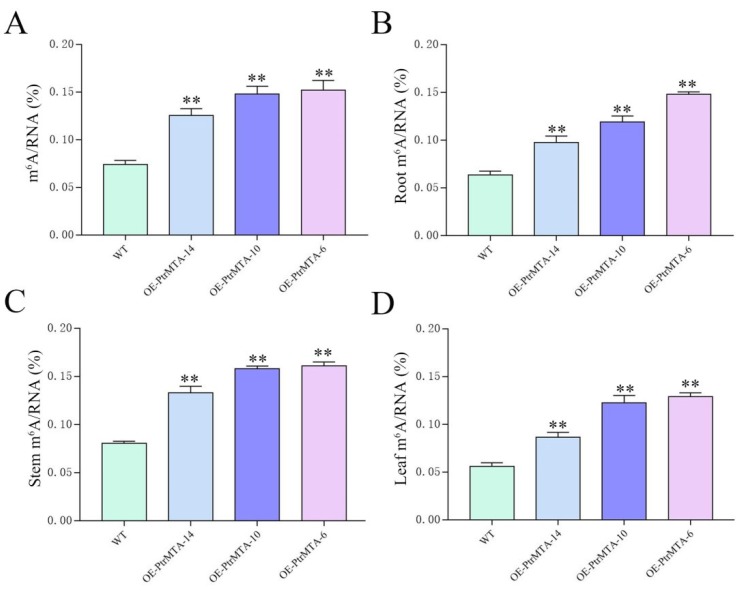
N6-methyladenosine (m^6^A) content in WT and *PtrMTA*-overexpressing plants under normal conditions. (**A**) m^6^A content in WT and *PtrMTA*-overexpressing plants at 30 days. (**B**) Analysis of the m^6^A content in the roots of WT and *PtrMTA*-overexpressing plants. (**C**) m^6^A content statistics in the stems of WT and *PtrMTA*-overexpressing plants. (**D**) Analysis of the m^6^A content in the leaves of WT and *PtrMTA*-overexpressing plants. Error bars are means ± SE (*n* = 6). All asterisks denote significant differences: ** *p* < 0.01.

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
