# Peer review of "MTA, an RNA m6A Methyltransferase, Enhances Drought Tolerance by Regulating the Development of Trichomes and Roots in Poplar"

_ijms, 2020, doi:10.3390/ijms21072462_

Round 1

Reviewer 1 Report

In the ms by Lu et al. the authors describe a Populus trichocarpa methyltransferase (MTA), overexpression of which in Poplar alters trichome density, root architecture. RNA methylation and enhances drought tolerance of the transgenic plants. The findings of the ms are interesting, but there are some throwbacks that need to be addressed before considering the ms for publication.

Figure 2: A control for nuclear localization (DAPI - 4′,6-diamidino-2-phenylindole) is missing. The authors should perform again the analysis, including this control. I am also al little concerned about the GFP and mCherry signals in the Figures. The signal is too diffused and if you compare it to the Arabidopsis homolog results (Vespa et al, Plant Physiology 2004 and Zhong et al. Plant Cell 2008), is much more different (nuclear specles). The authors should definitely pay attention in this analysis and include DAPI staining co-localization.

lines 134-144, Figure 3, Table S1: How did the authors made the measuremnts? Trichomes/leaf, trichomes/area? Please explain and correct the chart. What does trichomes statistics stand for? And on Table S1, what is statistical data? I can see from the images that the Oex plants have more trichomes, but I do not understand at all how the calculations were made and the statistical analysis. Please explain.

lines 169-170: Please add Day 20, on B and C in the Figure legend, just to be clear.

lines 176-179: How many nucleotides upstream of the ATG did the authors analyze? How many of each element did they recognize and which are the positions of these? The authors should make an extra Figure, additional to Table S2, were they should also add the number of each element recognized and its position relative to the ATG) showing all this information.

line 237: This is in Ref 13, Zhong et al., Plant Cell 2008. Please correct.

Line 247: "These results indicate that PtrMTA and PtrFIP37 may interact with each other": The authors either need to perform additional experiments to show that or revise this statement. With the present data you can only conclude that the two proteins co-localize. To state that you have to perform Y2H experiments, BiFC experiments and co-Immunoprecipitation experiments, to prove it.

line 279: "plants had strong drought resistance" -->plants were more resistant in drought stress, compared to WT plants. Please correct.

lines 288-289: " with supplemented with sunlight (16 h light and 8 h
289 dark)" : Is this sunlight or photoperiod controlled lights? What is the light intensity (umol photons/m-2/s-1)? Please define.

Please make a supplemental Table with all the primers used in this study and remove the sequences from the text)

Combine sections 4.2 and 4.5. The entire cloning procedure is better to be in one paragraph.

line 297: "cloned by PCR" : were the clones sequenced? How are the authors sure that there were no errors after 35 cycles of amplification? Which DNA polymerase did they use (Taq, Phusion? HiFi?)?

Paragraph 4.3: Which instrument did the authors use (model, manufacturer)? How was the relative expression analysis performed? Which method did they use (2-DCT, 2-DDCt)? Please explain.

Paragraph 4.4: Using which method and what "bait" were the homologues retrieved? BLASTp, BLASTn? Please explain. Please add a Supplemental Table with the GenBank IDs of the proteins used for analysis.

Paragraph 4.7: Which method did the authors use for Tobacco transient transformation? Also add DAPI staining co-localization as stated above.

line 358: "Nine OE-PtrMTA" --> 3 plants of the 3 transgenic lines. Please correct.

line 212: Compared to. Please correct.

line 105: 35:PrMTA --> 35S, please correct.

lines 123 and 125: please correct transected to transformed

Reviewer 2 Report

Dear Authors,

The manuscript by Lu et al. on “MTA, an RNA m6A methyltransferase, enhances drought tolerance by regulating the development of trichomes and roots in poplar” is a topic of interest for plant biologists, and is in the focus of IJMS. The paper deserves publication in IJMS. The results are very well described, commented and discussed. From my point of view as a biochemist and plant physiologist, this manuscript is valuable and should be accepted for printing.

Best regards and good health,

Referee

Round 2

Reviewer 1 Report

I am content with the changes made in the revised manuscript, so the ms can be accepted for publication in IJMS in the present form.

One short note, although the authors included the new Supplemental Tables generated in the ms pdf and they are fine, they were absent as standalone files in the "Supplemental files" file, so please do not forget to upload them at your final submission files.